complexity/e-science/systems theory

music charts, time scales, social acceleration, self-organized criticality

**Author for correspondence:**
Claudius Gros
e-mail: gros07@itp.uni-frankfurt.de

# Five decades of US, UK, German and Dutch music charts show that cultural processes are accelerating

## Lukas Schneider and Claudius Gros

Institute for Theoretical Physics, Goethe University Frankfurt, Frankfurt a. M., Germany

CG, 0000-0002-2126-0843

Analysing the timeline of US, UK, German and Dutch music charts, we find that the evolution of album lifetimes and of the size of weekly rank changes provide evidence for an acceleration of cultural processes. For most of the past five decades, number one albums needed more than a month to climb to the top, nowadays an album is in contrast top ranked either from the start, or not at all. Over the last three decades, the number of top-listed albums increased as a consequence from roughly a dozen per year, to about 40. The distribution of album lifetimes evolved during the last decades from a log-normal distribution to a power law, a profound change. Presenting an information–theoretical approach to human activities, we suggest that the fading relevance of personal time horizons may be causing this phenomenon. Furthermore, we find that sales and airplay-based charts differ statistically and that the inclusion of streaming affects chart diversity adversely. We point out in addition that opinion dynamics may accelerate not only in cultural domains, as found here, but also in other settings, in particular in politics, where it could have far reaching consequences.

## 1. Introduction

Music charts constitute a valuable source for the study of extended timelines of culturally and socially relevant data. One of the most influential collection of music charts, the US-based Billboard charts, has been used in this context to examine the evolution of popular music and to test theories of cultural change [1]. Other approaches concentrated on the fractional representation of race and gender [2,3], on the distribution of blockbusters among superstars [4], on linguistic and psychological aspects [5,6] and on the question whether

there is a trend towards a converging global popular music culture [7]. For the UK charts, a correlation analysis between musical trends, acoustic features and chart success has been performed [8]. On a general level, the interplay between significance and popularity has been investigated for the case of online music platforms [9].

An especially interesting aspect of music charts is that they allow to study if and how time scales that are potentially relevant for cultural and sociological developments have changed over the last five decades. This is a central theme for the theory of social acceleration [10], which presumes that social and cultural time scales have seen a continuing acceleration [11]. The pace of time is also a key determinant for liberal democracies [12], which are based on reliable temporal ties between politics and electorate [13].

Empirical studies attempting to determine quantitatively the long-term evolution of political, social or cultural time scales are rare [14], (H Rosa, personal communication). Here, we point out that music charts allow to investigate the long-term evolution of a given cultural time scale. For the US, the UK, the German and the Dutch charts, we find that several core chart characteristics, such as the overall chart diversity, the album lifetime and the entry position of number one albums, indicate that the pace of the underlying generative processes has accelerated substantially over the last decades, by a factor of two or more, in particular since the rise of the Internet. The evolution of the US Billboard and the German music charts are very similar, with the Dutch charts showing a time lag of roughly a decade. The UK charts are, on the other side, more conservative, in the sense that their statistics changed less dramatically since the early 1980s.

For number one albums, we find a complete reversal between the early decades, from the 1960s to the 1980s, and the situation as of today. In the past, essentially no number one album would start at the top of a chart. Reaching the top was instead a tedious climbing process that would take on the average an entire month, or more. Nowadays, the situation is the opposite. If an album is not the number one the first week of its listing, it has only a marginal chance to climb to the top later on. We believe that these empirical findings constitute quantitative evidence that the time scales determining cultural penetration and opinion formation processes have shortened substantially, in particular since the early 1990s.

Besides averaged quantities, we examine in detail the distribution of album lifetimes. The probability distribution that an album is listed overall for a certain number of weeks has seen a conspicuous evolution over the last three to four decades, with a log-normal distribution changing continuously into a power law. This evolution can be interpreted as a self-organizing process unfolding slowly over the course of several decades. This is a unique observation, as one can study, in general, only the dynamics of critical states, the end state, but not critical states in the making, viz. while they are forming [15].

The formation of log-normal and power-law distributions can be interpreted within an information–theoretical approach to human activities that we present. Within this approach, human activities are assumed to produce maximum entropy distributions, that is distributions for which the information content is maximal. The exponential distribution, which is entropy-maximal under the constraint of a given mean, becomes a power law once the Weber–Fechner Law is taken into account, namely that the brain discounts sensory stimuli, numbers and time logarithmically [16–18]. Considering next that people differ with respect to their preferences, which includes having distinct expectations for the mean of the distribution to be generated, one obtains a log-normal distribution. Using this framework, we propose that the observed change from a log-normal lifetime distribution to a power law is due to a decoupling of the individual time horizons from decision-making. There is no need to plan a trip to the next music store, to illustrate this statement, when an album can be bought on the spot, online, once somebody has discovered a song of her or his liking.

Music charts come in two varieties. As weekly sales charts, which is typically the case for albums, and as airplay charts, for which the number of times a song is aired by radio stations is counted. For airplay data, which are often included for single charts, the underlying generative process is the decision-making of a restricted number of radio operators. Sales statistics results in contrast from the collective behaviour of a potentially very large number of individuals. It is hence not surprising that the statistics of airplay and sales charts differs, as we find, on a fundamental level. The tendency to self-organize observed for sales charts does not manifest for airplay charts. In this study, we concentrate on sales, viz. album charts.

Within the last decade, most algorithms used to determine chart rankings have been updated with respect to the inclusion of streaming and downloads. We find that streaming tends to reduce both the number of albums making it to a chart, the chart diversity, and the inner mobility, that is the average weekly rank changes.

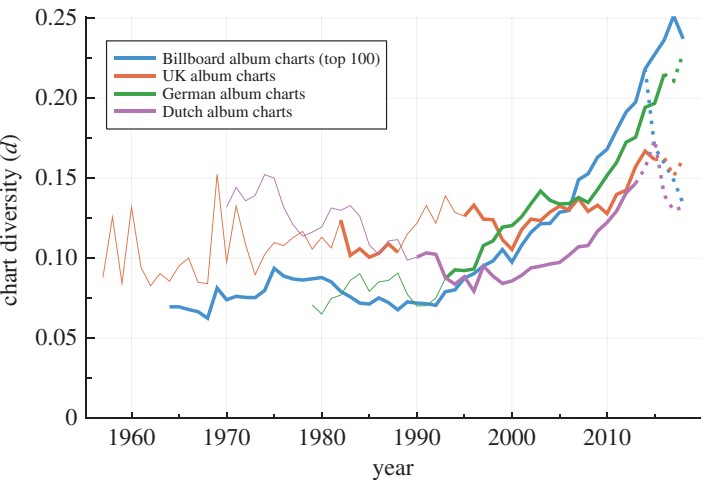

**Figure 1.** Chart diversity. The evolution of the chart diversity, which is defined as the fraction $d = N_a/N_s$ of the number of distinct albums $N_a$ listed in a given year and the number $N_s$ of slots available. Lines are thin for periods for which less than 100 chart positions are available, and dashed once streaming was included. For a top 100 chart and 52 weeks per year, there are $N_s = 100 \times 52$ slots. One observes that the chart diversity has increased steeply for the US and the German sales charts, in particular since 1990. For the Billboard album charts, the original sales-based rankings metric is available under a new name (full line), as of 2014/15, together with the updated version that is based on a multi-metric consumption rate (dashed line). The average number of weeks a song remains in the chart in a given year is of the order of $1/d$, compare figure 2.

## 2. Results

The US Billboard charts, the UK charts, the German and the Dutch music charts were obtained from public Internet sources [19–22]. An important parameter is the length of a chart, which typically increased over the years. For quantities that can be normalized with respect to the number of entries available in a given year, the entire timeline can be examined. For other features, quantities that depend on absolute and not on relative rankings, we restricted the analysis to charts that list at least the top 40/100 albums, which has been the case since 1963/1967/1978/1979, and, respectively, since 1963/1982/1993/1990, for the Billboard, the UK, the German and the Dutch album charts. Somewhat special are the US Billboard album charts, which increased to 200 in 1968. In the graphs we indicate, when suitable, whether top 100 or less chart ranks were available.

The algorithms used for the compilation of the individual charts have been adjusted over time, mostly in minor ways. A major update occurred for the Billboard charts in 2014/15, when the traditional sales-based ranking was substituted by a ranking based on a multi-metric consumption rate, which includes weighted song streaming. This update, which took effect at the end of 2014, affected the chart statistics profoundly. Streaming data were included since 2014/2016/2017 for the Dutch, the UK and the German charts, respectively.

The original Billboard album chart, the Billboard Top 200, was retained after the 2014/15 metric update under a new name, as 'Top Album Sales'. Data continuity is consequently achieved when using, as we have done, the Top Album Sales charts from 2014/15 on. Whenever possible, we will show the results obtained from both the Billboard Top 200 and the Top Album Sales charts, where the latter are compiled according to the unaltered sales-based ranking rules, albeit only for 100 ranks. For the UK, German and Dutch charts, only the version including streaming is accessible after the respective metric changes.

### 2.1. Chart diversity: the negative effect of streaming

For a first understanding of the data, we examine the evolution of the overall number of albums making it to the charts in a given year. As a gauge for the chart diversity, we normalize the number of distinct albums $N_a$ listed in a given year by the overall number $N_s$ of available slots. A top 100 chart could list, to give an example, a maximum of $N_s = 100 \times 52$ distinct albums per year. The average number of weeks $\bar{w}$ an album is listed in a given year is then just of the order of the inverse chart diversity, $\bar{w} \approx 1/d$. The overall chart lifetime of albums will be discussed further below.

In figure 1, the evolution of the chart diversity is presented on a year to year basis. One notes that the US and German charts follow qualitatively and quantitatively similar trends and that the chart diversity

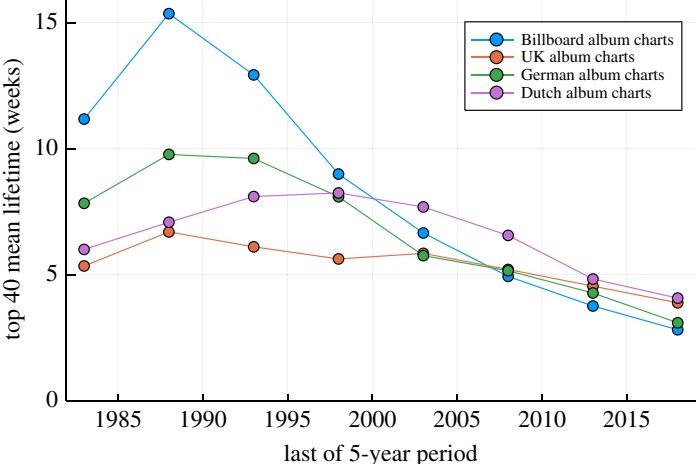

**Figure 2.** Album lifetime. The top 40 mean lifetime, namely the number of consecutive weeks an album is listed on the average among the top 40. The data have been pooled for trailing 5-year periods.

increases rapidly since the early 1990s. The average number of weeks $\bar{w}$ an album was listed in a given year in these two countries decreased correspondingly from about $1/0.075 \approx 13$ in 1990 to $1/0.2 = 5$ in 2014/15. For this instance, cultural processes were accelerated by a factor of more than two.

The Billboard charts data were split in 2014/15, when the traditional sales-based ranking was supplemented by a multi-metric consumption rate that includes streaming. For the latter, the trend to become more diverse reversed. Similar but less pronounced effects can be observed for the UK and the Dutch charts, but not for the German music charts. Note in this respect, that there are different routes, as detailed in the Appendix, on how to include streaming and song downloads.

Over their entire histories, the diversity of the British and Dutch music charts does not show pronounced trends. However, as visible in figure 1, a measurable increase in diversity is observed for the last two decades. The underlying reason for the otherwise distinct behaviours of the Dutch and UK charts with respect to the US and German charts is at present not evident.

## 2.2. Album lifetimes: a self-organized critical state in the making

For all charts, we have evaluated the number of weeks $n_w$ a given album remains within the listed range, the lifetime of an album. The lifetime is an absolute number which is not easily normalizable relative to the number of ranks available. To be able to compare the four charts over a comparatively long time span, we analyse only the top 40 albums. This restriction allows us to go back till 1979, the year when the Dutch charts increased their length to 40 ranks. For the Billboard charts, we checked also the long-term evolution for the top 100 albums, finding very similar trends.

The mean album lifetime pooled over trailing 5-year periods is presented in figure 2. The top 40 lifetime is roughly inversely proportional to the chart diversity shown in figure 1, which is, however, normalized to the chart length on a yearly basis. For the Billboard charts, the mean lifetime has seen a reduction by more than a factor two over the last 25 years.

Our main focus is the lifetime distribution, which is defined as the probability $P(n_W)$ for an album to remain listed $n_w$ weeks. We find that $P(n_W)$ can be fitted quite accurately by a log-normal distribution,

$$P(n_w) \sim e^{-a \ln (n_w) - b \ln^2 (n_w)}, \tag{2.1}$$

as shown as a log-log plot in figure 3 for the US Billboard charts. One observes that the lifetime distribution evolves steadily over the course of roughly four decades, from a quadratic to a linear dependency in a log-log representation. The lifetime distribution reduces to a power law $P(n_w) \sim 1/(n_w)^a$ in the limit $b \to 0$, an indication of a critical state.

In figure 4, the evolution of the fit parameters $a$ and $b$ entering (2.1) are shown for all charts investigated. One observes that $b$ tends to become small, in particular during the last 30 years, with the exponent $a$ approaching 2. We note that an exponent of 2 is marginal, as the mean of $P(n_w)$ diverges formally for $a < 2$ and $b = 0$. Nearly marginal exponents are, however, not uncommon, with a well-known example being the in-degree of domains in the World Wide Web [23].

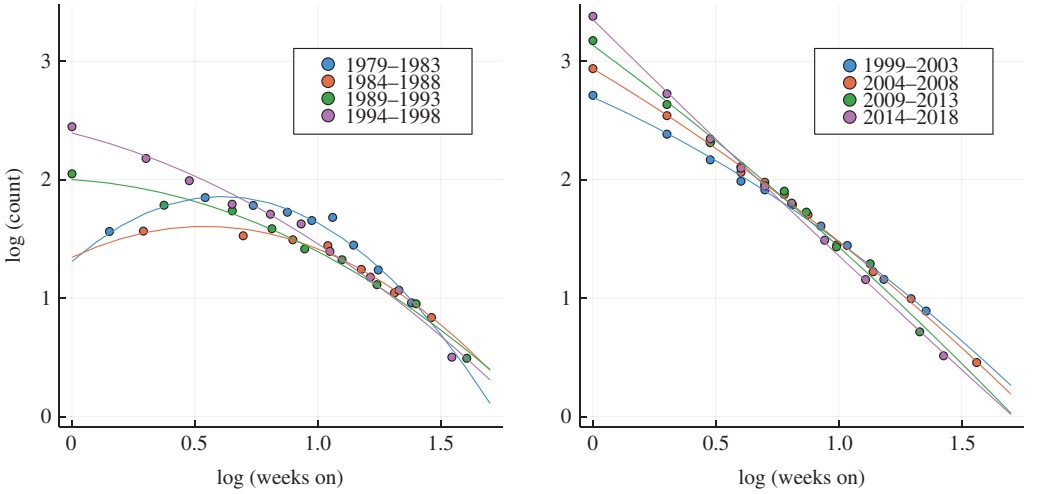

**Figure 3.** Lifetime distribution. On a basis 10 log-log plot, the top 40 lifetime distribution, namely the distribution of the number of weeks a given album is listed among the top 40 on the Billboard chart. The data (circles) have been pooled for successive 5-year periods and fitted quadratically (lines), compare (2.1). One observes that the lifetime distribution evolves over the years from a log-normal distribution towards a power law, which corresponds, respectively, to a quadratic and a linear dependency in a log-log representation.

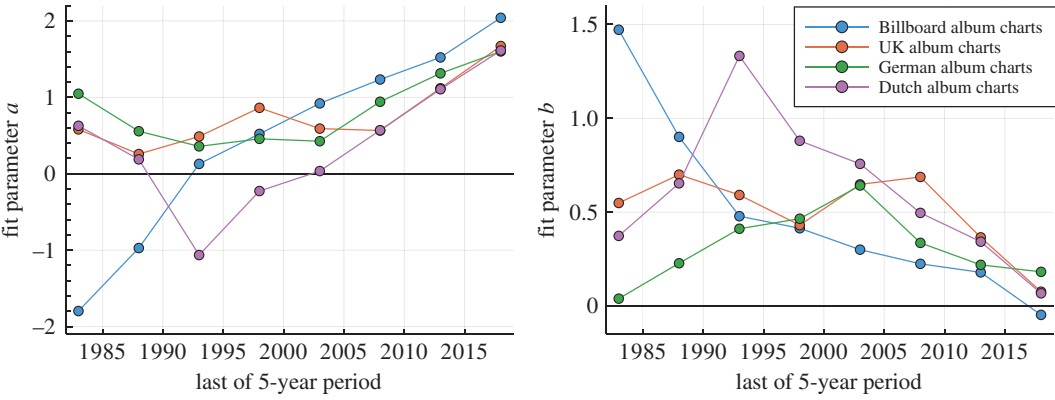

**Figure 4.** Coefficients of the lifetime distribution fits. The time evolution of the fit parameters $a$ (left panel) and $b$ (right panel) for the top 40 album lifetimes (circles), which correspond to the number of weeks $n_w$ an album is listed among the top 40. The lifetime distribution has been fitted by $\exp(-a \ln(n_w) - b \ln^2(n_w))$, as illustrated in figure 3 for the Billboard charts. This functional form corresponds to a log-normal distribution for $b > 0$ and to a power law for $b \to 0$. The data are for trailing 5-year periods. Over time, the lifetime distribution becomes more power law-like, with the exponent $a$ approaching the marginal value $a \to 2$. For the German music charts, the trend is less clear. The lines are guides to the eye.

The occurrence of power laws in evolving systems indicates the emergence of a self-organized critical state [24]. In general, it is to be expected that social systems, like the well-studied network of scientific collaborations [25], are characterized by evolving parameters. The data for the lifetime distribution presented in figures 3 and 4 are, however, unique, in the sense that it allows us to examine not only the final state, but the entire self-organizing process. A derivation of (2.1) based on an information–theoretical analysis will be presented in §3.

## 2.3. Number one albums: the start determines the fate

Commercial performance in terms of weekly sales varies vastly between albums. Of key importance is in this regard the first rank an album attains, the entry position. In figure 5, we present the probability that a number one album started as such. In the past, essentially no album started on the top, and albums succeeding to reach the top could take a month or more to do so. Today, the situation is reversed, with the reversal being nearly complete for the US, the German and the Dutch charts, and somewhat

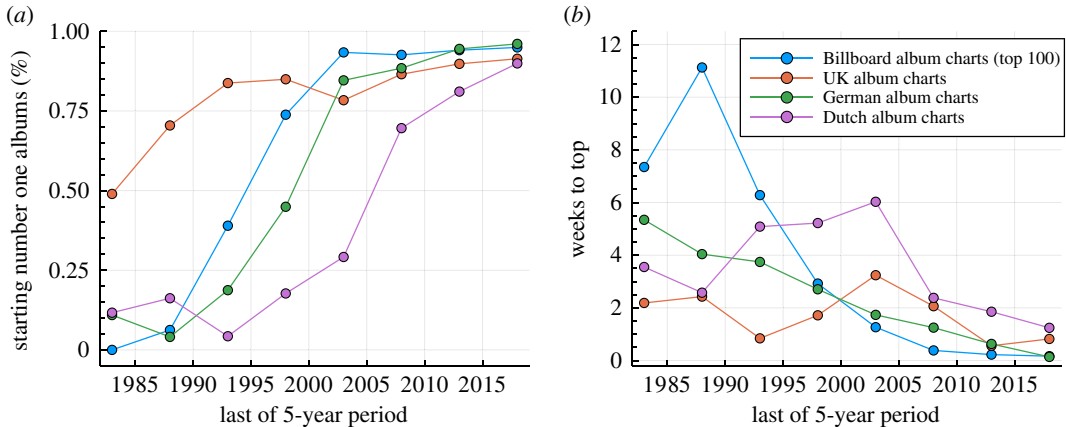

**Figure 5.** Number one albums. (*a*) The probability $P_{one}$ that a number one album started as such. The fraction of albums managing to reach the top when starting form a lower entry position is $1 - P_{one}$. (*b*) The average number of weeks number one albums needed to reach the top. Starting at the top corresponds to zero weeks.

reduced in magnitude for the UK charts. A time lag of about a decade is furthermore observable between the Billboard and the Dutch charts.

The rising predominance of number one entries is reflected in the number of weeks an album needs on the average to climb to the top, the climbing time. The zero for the data shown in figure 5 is set to the top, which implies that the climbing time for albums entering at the top is zero. It is quite remarkable that the average climbing time has seen, modulo fluctuations, a dramatic decrease for both the Billboard and the German music charts. This observation holds to a certain extend also for the Dutch charts, but not for the UK charts, which changed less over the last three decades. This more conservative evolution of the UK charts is consistent with the results for the chart diversity shown in figure 1. Overall, we believe that the data presented in figure 5 provide convincing evidence that the market penetration of new albums is now very fast, taking on the average a week or less, depending on the country. Three decades ago, the same process took about two to three weeks in the UK and more than the double in the US.

In table 1, we present for the Billboard album charts a compendium of statistical data that describes the dynamics of number one albums. For the order of the average first- and second-week ranks, one observes a reversal in the ordering. Before the mid-1990s, albums climbed; afterwards, the rank could only decay. Also evident is a substantial shortening of the number of weeks at the top, which was defined here as the number of weeks from the first to the last time an album was listed with a rank of one, hence including interruptions, which are of the order of 10–20%. As of today, albums are given little time to stay at the top, as number of albums attaining the top rank in a given year has increased so strongly, by a factor of 3–4 since the 1980s, that the number of yearly number one albums starts to approach the maximum of 52, compare table 1.

## 2.4. Entry and exit positions: ongoing symmetrization

How likely is it that a given album manages to climb at all, once in the charts? The evolution of the mean entry ranks is shown in figure 6 for the Billboard charts, together with the mean exit positions. For the full entry and exit distributions, also included in figure 6, Gaussian-broadened violin graphs have been generated, with the horizontal width being proportional to the probability of finding an entry/exit position within the respective five-year period. The data for the UK, German and Dutch charts are similar, but in part less pronounced.

The first-listing ranks are due to external effects, such as the quality of the album and the size and the penetration speed of publicity campaigns. The distance of the average exit ranks to the bottom, located at 100 in our case, is on the other side determined by the size of the average inner mobility, for which we will provide a specific definition in the next section. Here, we point out that the exit ranks are in general rising, which means that the inner mobility is accelerating.

The full entry rank distribution presented in figure 6 is consistent with the data for number one albums shown in figure 5, in the sense that the probability of higher entry positions has been continuously increasing since the early 1990s. A remarkable and somewhat astonishing result is the symmetry the entry distribution

**Table 1.** *Rank statistics of number one albums*. Rank statistics for albums is making it to the top. Data for the Billboard album charts, as averaged over trailing 5-year periods (given is, respectively, the final year). Shown are the means for the entry rank (#1 first), for the second week position (#1 second) and for the exit rank (#1 exit). Also given are the number of weeks to climb to the top (#1 climb), the time at the top (#1 top), including interruptions, the number of weeks from top to exit (#1 exit) and the average number of number one albums per year (#1 albums). For comparison, for all albums the mean for the entry and exits ranks (all first/all exit) are given, together with the average album lifetime (all lifetime). It is indicated whether the data are for the chart rank (R) or the number of weeks (W). Note the order reversal of the first- and second-week ranks of number one albums between 1998 and 2005. Compare figures 5 and 6.

| | | 1973 | 1978 | 1983 | 1988 | 1993 | 1998 | 2003 | 2008 | 2013 | 2018 |
|---|---|---|---|---|---|---|---|---|---|---|---|
| #1 first | R | 43.7 | 45.4 | 34.8 | 43.8 | 21.5 | 7.5 | 3.4 | 1.6 | 1.4 | 1.8 |
| #1 second | R | 17.6 | 22.1 | 20.2 | 26.3 | 10.1 | 6.3 | 5.0 | 4.4 | 5.0 | 8.1 |
| #1 exit | R | 91.7 | 86.3 | 89.5 | 93.6 | 92.2 | 93.6 | 91.9 | 89.9 | 85.1 | 74.9 |
| #1 climb | W | 5.4 | 7.1 | 7.2 | 11.1 | 6.3 | 2.9 | 1.3 | 0.4 | 0.2 | 0.2 |
| #1 top | W | 4.4 | 4.0 | 5.7 | 5.4 | 4.4 | 2.4 | 2.2 | 1.5 | 1.5 | 1.3 |
| #1 exit | W | 34.6 | 24.5 | 30.5 | 40.9 | 43.7 | 37.1 | 33.0 | 29.8 | 28.5 | 23.5 |
| #1 albums | | 12.4 | 14.0 | 10.0 | 10.8 | 13.2 | 22.6 | 25.6 | 36.0 | 35.0 | 41.0 |
| all first | R | 78.6 | 83.9 | 80.1 | 81.7 | 71.5 | 57.9 | 52.0 | 49.3 | 51.5 | 50.6 |
| all exit | R | 90.2 | 84.9 | 86.4 | 91.7 | 92.2 | 90.3 | 88.1 | 83.0 | 76.3 | 67.6 |
| all lifetime | W | 16.4 | 12.9 | 14.0 | 16.9 | 14.8 | 11.7 | 9.2 | 7.0 | 4.9 | 3.5 |

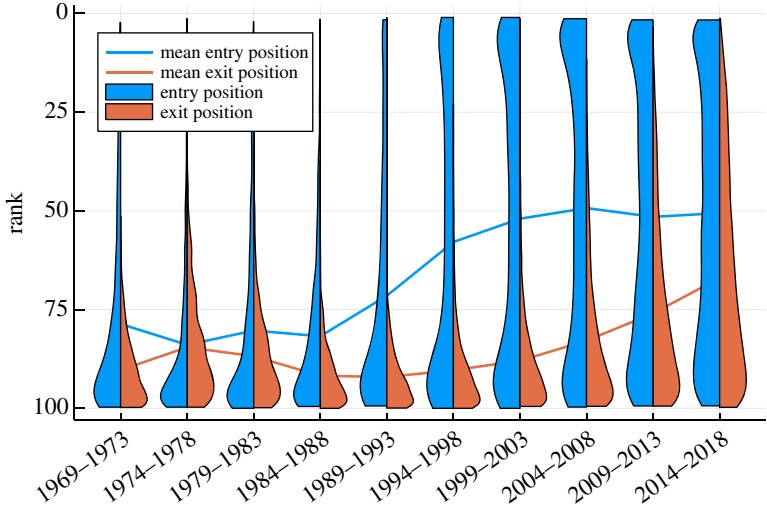

**Figure 6.** Entry and exit distributions. For the top 100 Billboard album charts, the distribution of entry (blue) and exit ranks (orange), averaged over five-year periods. The width of the violin charts measures the respective probabilities. Also included are the mean entry and exit positions (lines). Note the dramatic increase in top-ranked entries in the 1990s. The corresponding rise in exit ranks has been more sequential. Checking for different chart lengths, top 40 and top 200, we found a qualitatively similar behaviour that is rescaled according to the chart length considered.

exhibits nowadays with respect to 50, the half-way rank between the bottom and the top. We checked that this observation holds also for the entry distribution of top 40 and top 200 albums.

## 2.5. Inner mobility: accelerating rank decay

Once an album makes it to a chart, it may go up and down on a weekly basis. We define the relative inner mobility $M_I$ as

$$M_I = \left\langle \frac{R(t-1) - R(t)}{R} \right\rangle \quad \text{and} \quad R = \max(R(t), R(t-1)), \tag{2.2}$$

where $R(t)$ is the rank a given album has in week $t$, and where $\langle \cdot \rangle$ denotes the average over all albums within a certain period. Entry and exit weeks are not included. The max-function in (2.2) ensures that $|M_I| < 1$. For the sign, we have two cases.

— Climbing: $R(t) < R(t-1)$. The contribution to (2.2) is $(R(t-1) - R(t))/R(t-1)$, which is positive.
— Descending: $R(t) > R(t-1)$. The respective term is $(R(t-1) - R(t))/R(t)$, which is negative.

Instead of $M_I$, one can study the absolute inner mobility $R(t-1) - R(t)$, which would, however, weight an increase from 90 to 80 equal to an advancement from rank 11 to rank 1.

The data for the inner mobility presented in figure 7 shows that albums mostly lose rank on the average, namely that $M_I < 0$. It is also evident that the weekly rank loss tends to increase in size over time before streaming was included. Modulo substantial fluctuation, this is the case for all four charts investigated, with the German and the US charts trailing each other surprisingly closely. Similar downward trends are also observable for the Dutch and the UK charts, which was not the case for the chart diversity shown in figure 1. The timeline for the Dutch music chart can be interpreted by a time lag of roughly a decade. For the UK charts, $M_I$ decreased in contrast earlier, already during the 1970s and 1980s.

With $M_I$ being equivalent to a weekly decay rate, we can define the decay time $T_I \approx 1/M_I$, which measures the time scales of the inner dynamics. Between 1990 and 2010, $T_I$ increased by about a factor of three for the the Billboard, the German and the Dutch charts, with a similar acceleration happening for the UK charts 20 years earlier. In terms of the inner dynamics, all four charts indicate that cultural time has been accelerating, albeit not necessarily at the same time and at the same pace.

The observation that albums move down on the average implies that the average exit position is below the mean rank of first listings, which is consistent with the average entry and exit positions presented in figure 6. Recently, a class of model describing unidirectional growth processes that are terminated by a hard reset has been studied [26]. It is an interesting question whether the inner mobility, as presented in figure 7, could be described by an analogous but inverse dynamics.

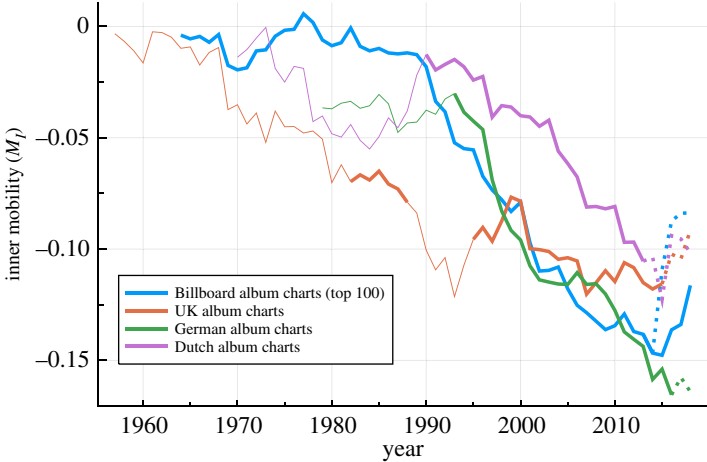

**Figure 7.** Inner mobility. On a year-by-year basis, the relative inner mobility $M_I$ of albums in the respective music charts, as defined by (2.2). Lines are thin for periods with less than 100 chart positions and dashed once streaming was included. Shown are the average weekly rank differences, in percentage. An $M_I$ of $-0.1$ corresponds to a rank decay rate of 10% per week, such as a decline from rank 9 to rank 10, or from 90 to 100. Entrance and exit weeks are not counted. For the Billboard album charts, a version retaining the original sales-based ranking metric remained available after the 20014/15 update, compare figure 1.

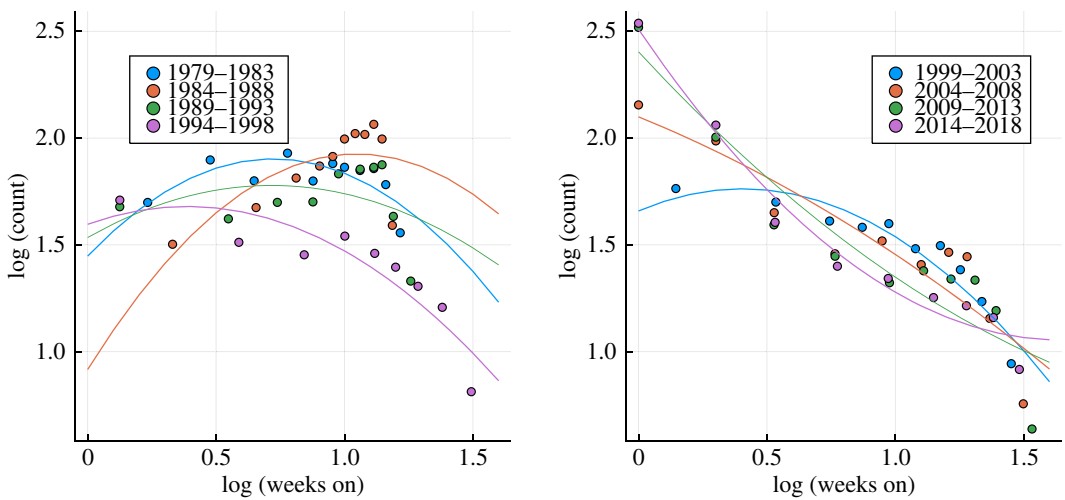

**Figure 8.** Single lifetimes. On a basis ten log-log plot, the top 40 lifetime distribution for the Billboard single charts, which are based in part on airplay data. Included are, as for figure 3, least-square quadratic fits, which correspond to log-normal and power-law distributions (2.1). It is evident that the single lifetime distribution cannot be approximated faithfully by log-normal distributions or power laws.

## 2.6. Billboard single charts: why airplay statistics differ

The Billboard single charts are based on a mix of sales data, jukebox playing and airplay, where the latter counts the number of times a song is aired by radio stations. The relative contributions have been adapted over the years, with a major change occurring in 2013, when streaming was included. We find that the Billboard album and single charts differ substantially with respect to their statistical properties, presumable because the single charts include airplay, whereas the album charts do not. Album sales are the result of a large number of individual decisions, whether to buy or not, which reflects an extended range of individual preferences. It is on the other side up to a relatively small group of radio operators to select the mix of songs that is likely to induce the targeted audience of the radio station to remain tuned in.

One can see that the lifetime distribution of singles and albums is distinct when comparing figures 3 and 8. The log-log plot of the top 40 single lifetimes presented in figure 8 shows that quadratic fits are very poor, which implies that single lifetimes cannot be described by (2.1) and that there is no evolution from a log-normal distribution to a power law. A certain tendency for the data to become

more straight is, however, present, possibly due to a crossover effect. Radio programme directors will be aware, in general, of the commercial success of the respective albums. Songs from top-ranked albums can hence be expected to enjoy a higher chance to be aired.

## 2.7. International statistical convergence

The picture emerging from our statistical analysis, like the inner mobility, see figure 7, is that the charts of two countries, Germany and the US, show very similar trends. This would be trivially the case if most of the songs making it to the charts in Germany and in the US were identical. A previous comparative study of American, Dutch, French and German popular music charts found, however, no evidence for an ongoing internationalization of popular music [7]. With regard to this question, which is not at the heart of our present investigation, we note that a comparison of the all-time most successful albums yields, as listed in table 2, a similar result. Among the all-time top five German albums two feature, to give an example, German 'Schlager', by Helene Fischer, with another one being a compilation of German action songs for the Kindergarten, 'Die 30 besten Spiel … '. There is, on the other side, more overlap between the all-time top 5 of the UK and the US album charts, even though these two charts differ to a certain extent with respect to their overall statistical properties. A more thorough investigation of this subject is left to future studies.

# 3. Information theory of human activities

A large-scale study of the statistics of data files publicly available on the Internet showed that the size distribution of formats having a time dimension, like videos and audio files, differs from static formats, such as jpeg and gif images [23]. The difference is that videos and audio files are log-normal distributed, with the file-size distribution of images following a power law. As a tentative explanation, it was suggested that the time domain corresponds for data files to a second dimension, in addition to resolution, and that the statistical distributions resulting from human activities may be analysed in many instances from an information–theoretical perspective [23]. Here, we suggest that the results presented in figures 3 and 4, namely that the statistics of album lifetimes evolved from a log-normal distribution to a power law, may be analysed along an analogous line of arguments.

Our underlying hypothesis is that there is a feedback loop between the activities carried out by a large number of individuals and the statistical ensembles produced by these activities. For the case of music charts, this presumption implies that there is a feedback between the lifetime distribution, which results from people buying music, and the individual decision to acquire a certain album. E.g. the decision to go for an album may be influenced by the number of weeks the album is already on the chart, and hence playing on the radio. If this hypothesis holds, it is reasonable to assume that the resulting distribution should maximize information in terms of Shannon's information entropy [27].

## 3.1. Human logarithmic discounting generates power laws

The neurophysiological processes that give rise to the ability of the human brain to process and record information determine a subjective value one attributes to an information source [16,17,28]. This relation is known as the Weber–Fechner Law. It states that the neural representations of sensory stimuli [16], numbers [17,29,30] and time [18,31] scale logarithmically, respectively, with the intensity of the bare stimulus, the number of objects and the length of a time span.

The Weber–Fechner Law determines which type of distribution, say of perceived stimuli $s$, is perceived to carry maximal information. Consider that the neural working regime prefers a certain mean for the perceived stimulus intensity, $\bar{s}$. The probability distribution $p(s)$ of perceived stimuli $s$ maximizing entropy is then an exponential, $p(s) \sim \exp(-s/\bar{s})$. The Weber–Fechner Law implies that the external, the physical measurable stimulus $S$, is logarithmically discounted, namely that $s = s_0 \ln(S)$, where $s_0$ is a characteristic scale. Using $p(s)\,\mathrm{d}s = p(S)\,\mathrm{d}S$, one then finds

$$p(S) = p(s)\frac{\mathrm{d}s}{\mathrm{d}S}, \quad p(S) \sim \frac{\mathrm{e}^{-s_0 \ln(S)/\bar{s}}}{S} \sim \frac{1}{S^a} \quad \text{and} \quad a = \frac{s_0}{\bar{s}} + 1, \tag{3.1}$$

for the maximal entropy distribution when expressed as a function of the afferent stimulus $S$. Maximization of information entropy under a logarithmic cost function hence yields generically a power law, as shown here for the case of a single relevant variable. This viewpoint is complementary

**Table 2.** The top five albums with the longest lifetimes (overall weeks on charts), for the Billboard charts (taking into account either the top 200 or the 100 ranks), and for UK, German and Dutch album charts. Listed is also the respective number of consecutive weeks and the achieved top rank, respectively from 1964/1957/1979/1970 until 2018.

| chart | artist | title | weeks (all) | weeks (con) | top rank |
|---|---|---|---|---|---|
| Billboard (200) | Pink Floyd | The Dark Side of the Moon | 939 | 593 | 1 |
| | Bob Marley and the Wailers | Legend: The Best Of … | 552 | 260 | 5 |
| | Journey | Journey's Greatest Hits | 542 | 156 | 10 |
| | Metallica | Metallica | 513 | 281 | 1 |
| | Guns N' Roses | Greatest Hits | 450 | 138 | 3 |
| Billboard (100) | Metallica | Metallica | 286 | 163 | 1 |
| | Adele | 21 | 260 | 144 | 1 |
| | Bob Marley and the Wailers | Legend: The Best of … | 247 | 31 | 5 |
| | Kendrick Lamar | Good Kid, m.A.A.d City | 236 | 56 | 2 |
| | Imagine Dragons | Night Visions | 232 | 112 | 2 |
| UK | ABBA | Gold - Greatest Hits | 865 | 125 | 1 |
| | Queen | Greatest Hits | 838 | 224 | 1 |
| | Bob Marley and the Wailers | Legend: The Best of … | 801 | 159 | 1 |
| | Fleetwood Mac | Rumours | 762 | 95 | 1 |
| | Meat Loaf | Bat Out of Hell | 520 | 203 | 9 |
| German | Andrea Berg | Best of | 352 | 269 | 18 |
| | Helene Fischer | Best of Helene Fischer | 337 | 301 | 2 |
| | ABBA | Gold - Greatest Hits | 317 | 60 | 1 |
| | S. S., K. G. & die Kita-Frösche | Die 30 besten Spiel- und B.-lieder | 290 | 33 | 43 |
| | Helene Fischer | Farbenspiel | 245 | 213 | 1 |
| Dutch | Adele | 21 | 330 | 167 | 1 |
| | André Hazes | De Hazes 100 | 303 | 196 | 2 |
| | Buena Vista Social Club | Buena Vista Social Club | 294 | 172 | 7 |
| | André Hazes | Al 15 jaar gewoon André | 257 | 58 | 3 |
| | Dire Straits | Brothers In Arms | 248 | 171 | 1 |

to dynamical approaches, such as the reinforcement loop via preferential attachment, that is the 'the rich get richer' principle [32]. For the case of music charts, one has in consequence that the lifetime distribution with the maximal information content is a power law.

## 3.2. Entropy maximization with variable mean

A maximum entropy distribution $\exp(-a\,s)$ is obtained by maximizing the objective function

$$\Phi(p) = -\int \mathrm{d}s\, p(s)\ln(p(s)) - a\int \mathrm{d}s\, s\, p(s), \qquad (3.2)$$

where $p(s)$ is the probability density of $s$. The first contribution to $\Phi(p)$ is the entropy and the second the weighted average $\bar{s} = \int \mathrm{d}s\, s\, p(s)$. The Lagrange parameter $a$ corresponds therefore to the relative weight of the average, the constraint. When $a$ is large, the constraint dominates maximization of $\Phi$.

We now assume that individuals differ with respect to how much importance they give to album lifetimes $s$, which will hence be reflected by the weight of the mean album lifetime $\bar{s}$. For this, we introduce a hidden variable $h$, such that individuals dispose of varying Lagrange parameters $a \to (a + \kappa h)$, where $\kappa$ is a coupling parameter. The joint distribution is then

$$p(s, h) \sim e^{-(a+\kappa h)s}p(h), \tag{3.3}$$

where $p(h)$ is the distribution of $h$, viz. the distribution of individual preferences. If we are interested only in the marginal distribution $p(s)$, which is typically the case when the hidden variable $h$ is not observable, as in our case, we obtain

$$p(s) \propto \int dh \, e^{-(a+\kappa h)s}p(h) \sim \int dh \, e^{-(a+\kappa h)s}e^{-(h-\bar{h})^2/(2\sigma_h^2)}, \tag{3.4}$$

when assuming that $h$ is normally distributed with mean $\bar{h}$ and standard deviation $\sigma_h$. We set $\bar{h} \to 0$ and absorb the mean $\bar{h}$ into the Lagrange multiplier $a$, which can be done without loss of generality. One finds that a Gaussian $p(h)$ leads to a Gaussian marginal $p(s)$,

$$p(s) \sim e^{-as-bs^2}, \quad b = \frac{(\kappa\sigma_h)^2}{2}, \quad s \propto \ln(S), \tag{3.5}$$

which turns into a log-normal distribution, see (2.1), once the Weber–Fechner log-discounting $s \propto \ln(S)$ is taken into account.

This result, equation (3.5), suggests that distributions of observables generated by the activity of a large number of individuals are log-normal when there is a substantial variability $\sigma_h$ of the perceived individual means $1/(a + \kappa h)$. One may view (3.5) as an alternative interpretation of the well-known result that Gaussians are maximum entropy distributions when both the first and the second moment, mean and variance, are given [27]. A maximum entropy distribution with an optimized mean and variance is hence equivalent to a maximum entropy distribution for which the optimized mean is variable.

The log-normal distribution (3.5) evolves into a power law for $b \to 0$, viz. in two cases, $\kappa \to 0$ and $\sigma_h \to 0$. The first case, $\kappa = 0$, implies that the hidden variable does not couple to the observable in the first place, having hence no effect. All individuals are identical in the second case, $\sigma_h = 0$. We note that $\Phi(p)$ is a functional of $p(s)$, which implies $\Phi(p)$ acts as a generating functional, akin to the role generating functionals take in the context of guided self-organization [33,34], such as for attractor relict networks [35] and Hebbian learning rules [36].

## 3.3. Time horizons are less important when time accelerates

People differ substantially in behavioural relevant traits, such as the perception of time [37]. The observation that individual likings are caused, in general, by a multitude of factors [38] suggests that the distribution of preferences can be approximated by a Gaussian and that (3.4) constitutes a faithful representation of a maximum entropy distribution when individual expectations vary.

For the case of music charts, we concentrated on the album lifetime as our primary observable. Chart rankings and the lifetime are determined by weekly buying decisions that depend not only on a range of external factors, such as prominent marketing campaigns, but also on the performance of albums on the chart. We postulate here that the hidden variable entering the information–theoretical interpretation via (3.4) is related to the individual perception of time, the time horizon. A log-normal distribution would then be observed when a substantial coupling $\kappa$ to the individual time horizons is present. In this case, it would matter, for a buying decision, how long the album in question has already been listed, and hence aired by radio stations and on the Internet. A power law is recovered on the other side when the coupling to the individual time horizons ceases to be relevant.

The presumption of a decreasing relevance of the time domain implies that one-time effects suffice increasingly to influence buying decisions. This scenario is not unlikely, given that the rise of the Internet opened the possibility to buy music essentially on the spot, e.g. directly after one has heard or discovered a song, online or on the radio. There is no need to plan for a trip to the next music store on a free afternoon, an undertaking that easily led in the past to delays of days and weeks, and with this to a coupling between buying and the personal time horizon. We stress, however, that the arguments for a progressive decoupling of the time horizon are at present only circumstantial and that we cannot rule out that other drivings may cause the observed changes in the chart statistics.

Our argument that personal time horizons may have seen a decoupling from buying decisions draws support also from the increasing relevance of top entry ranks, as shown in figure 5 for number one

albums. Before 1990, only very few albums succeeded to enter the charts as number one, which implies that publicity and commercial success needed time, several weeks at least. The situation has changed since the advent of the Internet, which allows news about new releases to spread very fast via social media channels. For most people, buying an album will not affect the monthly budget substantially, which is hence a decision that can be carried out without further evaluation once taken. The influence of the time domain on buying albums is hence reduced.

## 4. Discussion: political time scales and stability

One may ask whether the acceleration of the information flow observed here for the case of music charts may be the reverberation of an equivalent speed-up of societal and political processes at large. In this regard, we point out that democracies rely in general on a stable and continuous evolution of public opinion [39] and that it has been suggested that not only the content of the political discourse is what matters, but in particular also the speed at which public opinion changes [40].

Cultural and political processes condition each other [41], which implies that the respective time scales couple [42] and that social acceleration will induce, if present and ongoing, a growing mismatch between political time delays, which are entrenched in a representative democracy by the electoral cycle, and the accelerating pace of opinion dynamics. This is an observation with potentially far reaching consequences, as it is known from dynamical systems theory that a mismatch of instantaneous and delayed feedback induces instabilities [27]. The outlook is then, from a dynamical systems perspective, that modern democracies become inevitably unstable once the time scale of public opinion formation is shorter than the time delays characterizing the interaction between electorate and political decision-making [40]. Whether democracy as such is already in crisis is a question of debate [43–45].

## 5. Conclusion

Book, music and other charts are compiled in order to satisfy the unabated interest [46,47] in the commercial and artistic success of music albums, as well as in other products of popular and classical culture. They provide a valuable source for long-term socio-artistic studies as their fundamental ranking criterion, success, has not changed over the last 50 years, albeit modulo technical adjustments. Given the continuity of the ranking metric, changes of the chart statistics occurring over the time span of several decades are, therefore, reflecting long-term socio-cultural developments.

We find three major trends. Firstly, one observes a substantial increase in the overall number of albums making it to the chart on a yearly basis, the chart diversity. The number of number one albums increased even more strongly; it is nowadays around 40 per year for the Billboard album charts. Secondly, the route to become a number one hit has changed dramatically. Instead of climbing arduously from a modest entry rank, number one hits start nowadays as such. Finally, we observe that the statistics of album lifetimes has seen a conspicuous change, evolving over the course of several decades towards a critical state. To our knowledge, this is the first instance that the self-organization process as such may be studied explicitly [27]. Within a proposed information–theoretical approach to human activities, the resulting power-law distribution of album lifetimes is due to the growing irrelevance of individuality, in the sense that the time necessary to form an opinion on whether to acquire an album, and to buy it, is now very low. It does not matter if somebody needs only a few minutes to decide to download an album, or as long as a few days, as both time scales are below the chart frequency of one week.

From an additional angle, one can interpret the acceleration of chart processes found in this study as a measurable indication that the cultural and social exchange of information occurs nowadays at a substantially faster rate than it used to. While intuitive, this observation could imply that the pace of opinion formation may have accelerated likewise over the past five decades. This would be a worrisome result, as it has been reported that representative democracies need to deal with the growing mismatch between the time delays inherent in political decision-making and an ever faster opinion dynamics [40], or face an uncertain future.

Data accessibility. The paper deals with an analysis of music charts. The source of the data is given explicitly [19–22]. Readers can access the data source, as it is publicly available.

Authors' contributions. Data acquisition and primary analysis by L.S., study concept and interpretation by C.G.

Competing interests. We declare we have no competing interests.

Funding. No funding has been received for the article.

Acknowledgements. C.G. thanks Nathan Valentí for discussions.

# Appendix A

In table 2, the top five most successful albums are listed, according to the overall lifetime, the number of weeks on the respective charts. For the US Billboard charts which have a length of 200, the album lifetimes have been evaluated taking into account either 100 or 200 ranks. For the UK, the German and the Dutch album charts, the respective entire data have been evaluated. Also given is the number of consecutive weeks and the respective highest rank reached.

Complementing the major rule updates given in §2, we present here for completeness an extended history of the Billboard album charts [48,49].

1963: The Billboard album charts start with top 150.

1967: Extension first to 175, and then to 200.

1991: Data source changed from phone call sampling of record stores to 'Nielsen SoundScan'.

2010: Catalogue albums (older than 18 months, rank below 100, no single) are included. Previously they were dropped.

2014: Metrics changed from a sales-based ranking to one measuring multi-metric consumption, which includes streaming. Sales-based album chart takes the name 'Top Album Sales'.

The weighting of streaming can be performed along several routes. When introduced for the Billboard album charts, 10 song sales or 1500 song streams from an album were treated as equivalent to one purchase of the album. This changed in 2018, when 1250 premium audio streams, 3750 ad-supported streams, or 3750 video streams were consider to equal one album unit. For the UK charts, only the 12 most streamed songs of an album contribute instead, and not all. The weekly revenue and not the number of downloads enter on the other side the German charts.

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
