## [Reviewer comments · Royal Society Open Science]

Review History

RSOS-190567.R0 (Original submission)

Review form: Reviewer 1 (Zoltan Neda)

Is the manuscript scientifically sound in its present form?

Yes

Are the interpretations and conclusions justified by the results?

Yes

Is the language acceptable?

Yes

Is it clear how to access all supporting data?

Yes

Do you have any ethical concerns with this paper?

No

Have you any concerns about statistical analyses in this paper?

No

Recommendation?

Accept with minor revision (please list in comments)

Comments to the Author(s)

The work presents evidences for the changes that are induced by an accelerated selection process over the dynamics in the top-list of music charts (Billboard, UK, German and Dutch). Convincing data is given for visualizing the difference in the top-list dynamics and statistics of music charts between 1960-90 and 1990-2018. The accelerated rhythm is nicely visible in the chart diversity evolution. The distribution time for the time periods in which an album persist in the top 40 positions has also visible differences between these two time-periods. While in the period 1960-1990 one observes a lognormal distribution, in the 1990-2018 period a scaling is observable (power-law trend). The authors analyze also the differences in the evolutionary trends of an album in the top-list. In the period 1960-90 mostly an increasing trend is dominating (albums start at lower positions and climb towards the top....), while in the 1990-2018 period a decreasing trend is observable (albums start right in the top and slowly descends in the list). In contrast with albums, single songs do not have such a clear trends and statistics. The findings are explained in the context of information-theoretical approaches using entropy maximizing principles and the Weber-Fechner law. The manuscript is clearly presented and the topic is of broad interest. It is definitely a work that should be published, however before publication I would suggest the authors to take into account the following recommendations:

1. Why the authors focus only on the Western world. I assume that Asia is nowadays a leading market. Are there long-term charts available also for Japan, South Korea, etc...? Maybe some comments in this sense would help us.
2. The notation "log" is used for logarithm throughout the manuscript, without specifying if it is a ten based or "e" based logarithm. In some graphs, I had the feeling it is a ten based one, however in the equations I would tend that it is an "e" based one. Please use either the accepted notations: "lg" and "ln" or specify it.
3. The term "unique album" could be a little confusing and not necessarily pointing to what the authors mean by that. Maybe some other expression or some explanations would be OK.
4. Equation 3.1 is incorrect. the P(S) scaling assuming the Weber-Fechner law should be $P(S)=1/S^a$ with $a=s_0/s_{bar}-1$.
5. In Equation 3.2 is not totally convincing for me the manner in which the coupling is taken into account. I do understand that one needs a coupling between the "h" and "s" variables, but why in the exponent (?) only for the sake of analytical simplicity?

Review form: Reviewer 2**Is the manuscript scientifically sound in its present form?**

Yes

Are the interpretations and conclusions justified by the results?

No

Is the language acceptable?

Yes

Is it clear how to access all supporting data?

No

Do you have any ethical concerns with this paper?

No

Have you any concerns about statistical analyses in this paper?

I do not feel qualified to assess the statistics

Recommendation?

Accept with minor revision (please list in comments)

Comments to the Author(s)

I overall liked this paper and I thought the data was interesting and the modelling in section 3 was good. My two main quibbles:

1. The term "cultural time" implies to me that there is some underlying phenomenon that is making all cultural things accelerate at about the same rate. This seems pretty unlikely to me, but I guess you could demonstrate it by looking at data from many different cultural domains. The paper only looks at one cultural domain. And even if many domains were considered, wouldn't it just be straightforward to say many cultural processes are speeding up. That this is even true seems to be belied by Figure 1 where cultural time was accelerating until 2014, and then switched to decelerating. I know that what was measured changed in that year, but that is my point: if there was a "cultural time" that was accelerating, it wouldn't matter what we measured, we should still see it.

2. Given my skepticism about cultural time, I don't see what relevance the paper has to politics. But politics appear in the first sentence of the abstract. If you want to claim that political opinions are changing faster, then I think you really just have to look at political opinions.

So in my opinion the paper would be improved by deleting "cultural time" and downplaying politics as a motivation.

smaller comments/questions:

page 1. in the abstract you use the phrase "unique phenomenon". How do you know this is a unique phenomenon? Something going from a lognormal distribution to a power law distribution has never happened before?

page 2 "a unique observation" Again, can you say a little more about this? It's never been observed before in cultural data? Or any kind of data? What typically happens then.

page 2 sentence beginning with "It is postulated that human activities...." was hard to understand. Maybe delete "to" after "viz".

page 4. "1990th" -> "1990s"

in lines 56 and 57 replace 1.0 with 1. (When I see 1.0 I assumed it was from data)

page 5. "The trend to become more diverse was dramatically reversed..." I don't think any real thing reversed trend. You just switched from looking at one thing that was increasing to another thing that was decreasing. It might be better just not to include the latter data.

page 6. "raising prominence" -> "rising prominence"

page 11. "give raise" -> "give rise"

page 13 "90th" -> "90s"

Review form: Reviewer 3**Is the manuscript scientifically sound in its present form?**

Yes

Are the interpretations and conclusions justified by the results?

No

Is the language acceptable?

Yes

Is it clear how to access all supporting data?

Yes

Do you have any ethical concerns with this paper?

No

Have you any concerns about statistical analyses in this paper?

No

Recommendation?

Major revision is needed (please make suggestions in comments)

Comments to the Author(s)

The paper "Five decades of US, UK, German and Dutch music charts show that cultural time is accelerating" provides an analysis of the song dynamics on top charts. Several interesting trends are found, including an accelerated turnover of songs on charts, the increased likelihood of songs entering as #1, and a change in the songs' lifetime distribution. I think this is an interesting study, but I find some of the interpretations made by the authors problematic. Also, below I give comments on how the analysis and presentation can be made clearer.

The main claim of the authors is that the observed statistical results that they obtain inform us about the changes in cultural time. Perhaps the authors should explain exactly what this means. The changes they observe are intuitively explained by the advent of the internet and information accessibility. Is this what they mean by cultural time? It is clear that information has become much more accessible across all cultural domains, and a detailed chart study is not necessary to claim this. Therefore, I would not try to interpret the findings in the general "cultural time" framework, but rather concentrate on the identifying the role of the technology on chart behavior and music consumption. I would not interpret the results as significant for democracy, politics, and other unrelated things. It is far fetched and distracts the reader from the interesting results of the paper.

Further, I think a more in-depth study of the chart rules should be presented. There have been a lot of changes in the ways the charts were formed, especially in the 90s and 2000s. These changes will obviously affect the observables such as long lifespan, entry ranks, etc. This information should be presented and used to explain the data.

Below are detailed comments.

Abstract: The abstract starts with statements about electoral cycles and modern democracies, but this seems somewhat removed from the main topic of the research. I suggest removing any mention of politics and motivate the research in a more direct way, by talking about the time scales if music preference changes.

Page 3 line 20 - 25: connecting some results from dynamics systems with political instabilities is very far fetched. It is not clear why these particular results should be applicable. At this point in the paper we don't know what the observations are exactly, and further it is unclear why an instability in a dynamics system has any relevance for "democracy in crisis". I suggest to either explain exactly how your results are connected with political instabilities (not in the intro, but in the discussion, after we know what the results are), or remove this part completely.

Page 3 lines 42-44: "We believe that these empirical findings constitute quantitative evidence that the time scales determining cultural penetration and opinion formation processes have shortened substantially, in particular since the early 90th." What is cultural penetration? Also, how do we know that the observations are caused by the shortening of the opinion formation, and not by any number of other reasons, much more mundane (e.g. the way the charts are created, the way music is consumed these days compared to the past, etc)?

Page 3, line 48: what does "a unique observation" refer to?

Page 3, line 54: what are "sensori stimuli" (a typo?) In general, this whole paragraph is not clear. It explains results that have not been presented yet, and it does so in a way that is not accessible to people not working directly in this type of problems. I suggest to move this to the discussion and provide more in-depth explanations.

Page 4, sales vs airplay, e.g. line 48: "In this study we concentrate mostly on sales, viz album charts." It appears that US top 100 charts contain information on both sales and airplay; please clarify.

Page 6, line 34: "Note in this respect, that there are different routes on how to include streaming and song downloads" -- please clarify what this means.

Page 6, line 56: please remove "astonishing", and use a more quantitative description of the goodness of fit.

Page 7, line 55: "has been normalized such that the climbing time for albums entering at the top is zero" -- please explain this normalization procedure.

Page 8, figure 5: It could be informative in addition to show the probability distribution (a histogram) of the initial rank of songs that ended up as #1, for example, for 1985 and for 2015.

Pages 7-8: Number one entries. From Wikipedia: "During the 1990s, <...> accusations began to fly of chart manipulation as labels would hold off on releasing a single until airplay was at its absolute peak, thus prompting a top ten or, in some cases, a number one debut. In many cases, a label would delete a single from its catalog after only one week, thus allowing the song to enter the Hot 100, make a high debut and then slowly decline in position as the one-time production of the retail single sold out." -- can this have something to do with an increased tendency to debut at number one position?

Page 8: Exit ranks. There is a rule (in Billboards charts) that albums cannot remain on the charts for more than 20 weeks (they are taken off the charts). This will influence the exit ranks.

Page 11 figure 8: Even though the fits are not as good as in figure 3, the tendency is very similar: the lines straighten out from being parabola-like. Should one interpret the worse fit as a qualitative, fundamental difference, or a quantitative one?

Page 12-13, sections (b) and (c) : what is the meaning of the hidden variable, h , in the context of music charts? In (c) it is stated that this is individual time horizons, but it is not very clear what this means. Please clarify this connection, otherwise the derivation in (b) does not apply.

Page 15, section 4, last paragraph: again, I think the relevance of this study to politics is not clear.

Decision letter (RSOS-190567.R0)

15-May-2019

Dear Dr Gros:

Manuscript ID RSOS-190567 entitled "Five decades of US, UK, German and Dutch music charts show that cultural time is accelerating" which you submitted to Royal Society Open Science, has been reviewed. The comments from reviewers are included at the bottom of this letter.

In view of the criticisms of the reviewers, the manuscript has been rejected in its current form. However, a new manuscript may be submitted which takes into consideration these comments.

Please note that resubmitting your manuscript does not guarantee eventual acceptance, and that your resubmission will be subject to peer review before a decision is made.

Your resubmitted manuscript should be submitted by 12-Nov-2019. If you are unable to submit by this date please contact the Editorial Office.

on behalf of Prof Marta Kwiatkowska (Subject Editor)
openscience@royalsociety.org

Associate Editor Comments to Author:

It is perplexing that the authors have submitted to Royal Society Open Science a manuscript that is evidently in the cultural evolution space. This is not a comment on the paper's quality (indeed, the reviewers have provided extensive queries on this front), but rather a comment on whether the authors have selected the correct journal for their work. If the authors are able to satisfy the reviewers that their required changes have been made, we will certainly consider the paper for publication, but the authors might consider a more field-specific journal for other papers in this realm.

Reviewers' Comments to Author:

Reviewer: 1

Comments to the Author(s)

The work presents evidences for the changes that are induced by an accelerated selection process over the dynamics in the top-list of music charts (Billboard, UK, German and Dutch). Convincing data is given for visualizing the difference in the top-list dynamics and statistics of music charts between 1960-90 and 1990-2018. The accelerated rhythm is nicely visible in the chart diversity evolution. The distribution time for the time periods in which an album persist in the top 40 positions has also visible differences between these two time-periods. While in the period 1960-1990 one observes a lognormal distribution, in the 1990-2018 period a scaling is observable (power-law trend). The authors analyze also the differences in the evolutionary trends of an album in the top-list. In the period 1960-90 mostly an increasing trend is dominating (albums start at lower positions and climb towards the top...), while in the 1990-2018 period a decreasing trend is observable (albums start right in the top and slowly descends in the list). In contrast with albums, single songs do not have such a clear trends and statistics. The findings are explained in the context of information-theoretical approaches using entropy maximizing principles and the Weber-Fechner law. The manuscript is clearly presented and the topic is of broad interest. It is definitely a work that should be published, however before publication I would suggest the authors to take into account the following recommendations:

1. Why the authors focus only on the Western world. I assume that Asia is nowadays a leading market. Are there long-term charts available also for Japan, South Korea, etc...? Maybe some comments in this sense would help us.
2. The notation "log" is used for logarithm throughout the manuscript, without specifying if it is a ten based or "e" based logarithm. In some graphs, I had the feeling it is a ten based one, however in the equations I would tend that it is an "e" based one. Please use either the accepted notations: "lg" and "ln" or specify it.
3. The term "unique album" could be a little confusing and not necessarily pointing to what the authors mean by that. Maybe some other expression or some explanations would be OK.
4. Equation 3.1 is incorrect. the $P(S)$ scaling assuming the Weber-Fechner law should be $P(S)=1/S^a$ with $a=s_0/s_{bar}-1$.
5. In Equation 3.2 is not totally convincing for me the manner in which the coupling is taken into account. I do understand that one needs a coupling between the "h" and "s" variables, but why in the exponent (?) only for the sake of analytical simplicity?

Reviewer: 2

Comments to the Author(s)

I overall liked this paper and I thought the data was interesting and the modelling in section 3 was good. My two main quibbles:

1. The term "cultural time" implies to me that there is some underlying phenomenon that is making all cultural things accelerate at about the same rate. This seems pretty unlikely to me, but I guess you could demonstrate it by looking at data from many different cultural domains. The paper only looks at one cultural domain. And even if many domains were considered, wouldn't it just be straightforward to say many cultural processes are speeding up. That this is even true seems to be belied by Figure 1 where cultural time was accelerating until 2014, and then switched to decelerating. I know that what was measured changed in that year, but that is my point: if there was a "cultural time" that was accelerating, it wouldn't matter what we measured, we should still see it.
2. Given my skepticism about cultural time, I don't see what relevance the paper has to politics. But politics appear in the first sentence of the abstract. If you want to claim that political opinions are changing faster, then I think you really just have to look at political opinions.

So in my opinion the paper would be improved by deleting "cultural time" and downplaying politics as a motivation.

smaller comments/questions:

page 1. in the abstract you use the phrase "unique phenomenon". How do you know this is a unique phenomenon? Something going from a lognormal distribution to a power law distribution has never happened before?

page 2 "a unique observation" Again, can you say a little more about this? It's never been observed before in cultural data? Or any kind of data? What typically happens then.

page 2 sentence beginning with "It is postulated that human activities...." was hard to understand. Maybe delete "to" after "viz".

page 4. "1990th" -> "1990s"

in lines 56 and 57 replace 1.0 with 1. (When I see 1.0 I assumed it was from data)

page 5. "The trend to become more diverse was dramatically reversed..." I don't think any real thing reversed trend. You just switched from looking at one thing that was increasing to another thing that was decreasing. It might be better just not to include the latter data.

page 6. "raising prominence" -> "rising prominence"

page 11. "give raise" -> "give rise"

page 13 "90th" -> "90s"

Reviewer: 3

Comments to the Author(s)

The paper "Five decades of US, UK, German and Dutch music charts show that cultural time is accelerating" provides an analysis of the song dynamics on top charts. Several interesting trends are found, including an accelerated turnover of songs on charts, the increased likelihood of songs entering as #1, and a change in the songs' lifetime distribution. I think this is an interesting study, but I find some of the interpretations made by the authors problematic. Also, below I give comments on how the analysis and presentation can be made clearer.

The main claim of the authors is that the observed statistical results that they obtain inform us about the changes in cultural time. Perhaps the authors should explain exactly what this means. The changes they observe are intuitively explained by the advent of the internet and information accessibility. Is this what they mean by cultural time? It is clear that information has become much more accessible across all cultural domains, and a detailed chart study is not necessary to claim this. Therefore, I would not try to interpret the findings in the general "cultural time" framework, but rather concentrate on the identifying the role of the technology on chart behavior and music consumption. I would not interpret the results as significant for democracy, politics, and other unrelated things. It is far fetched and distracts the reader from the interesting results of the paper.

Further, I think a more in-depth study of the chart rules should be presented. There have been a lot of changes in the ways the charts were formed, especially in the 90s and 2000s. These changes will obviously affect the observables such as long's lifespan, entry ranks, etc. This information should be presented and used to explain the data.

Below are detailed comments.

Abstract: The abstract starts with statements about electoral cycles and modern democracies, but this seems somewhat removed from the main topic of the research. I suggest removing any mention of politics and motivate the research in a more direct way, by talking about the time scales if music preference changes.

Page 3 line 20 - 25: connecting some results from dynamics systems with political instabilities is very far fetched. It is not clear why these particular results should be applicable. At this point in the paper we don't know what the observations are exactly, and further it is unclear why an instability in a dynamics system has any relevance for "democracy in crisis". I suggest to either explain exactly how your results are connected with political instabilities (not in the intro, but in the discussion, after we know what the results are), or remove this part completely.

Page 3 lines 42-44: "We believe that these empirical findings constitute quantitative evidence that the time scales determining cultural penetration and opinion formation processes have shortened substantially, in particular since the early 90th." What is cultural penetration? Also, how do we know that the observations are caused by the shortening of the opinion formation, and not by any number of other reasons, much more mundane (e.g. the way the charts are created, the way music is consumed these days compared to the past, etc)?

Page 3, line 48: what does "a unique observation" refer to?

Page 3, line 54: what are "sensori stimuli" (a typo?) In general, this whole paragraph is not clear. It explains results that have not been presented yet, and it does so in a way that is not accessible to people not working directly in this type of problems. I suggest to move this to the discussion and provide more in-depth explanations.

Page 4, sales vs airplay, e.g. line 48: "In this study we concentrate mostly on sales, viz album charts." It appears that US top 100 charts contain information on both sales and airplay; please clarify.

Page 6, line 34: "Note in this respect, that there are different routes on how to include streaming and song downloads" -- please clarify what this means.

Page 6, line 56: please remove "astonishing", and use a more quantitative description of the goodness of fit.

Page 7, line 55: "has been normalized such that the climbing time for albums entering at the top is zero" -- please explain this normalization procedure.

Page 8, figure 5: It could be informative in addition to show the probability distribution (a histogram) of the initial rank of songs that ended up as #1, for example, for 1985 and for 2015.

Pages 7-8: Number one entries. From Wikipedia: "During the 1990s, <...> accusations began to fly of chart manipulation as labels would hold off on releasing a single until airplay was at its absolute peak, thus prompting a top ten or, in some cases, a number one debut. In many cases, a label would delete a single from its catalog after only one week, thus allowing the song to enter the Hot 100, make a high debut and then slowly decline in position as the one-time production of the retail single sold out." -- can this have something to do with an increased tendency to debut at number one position?

Page 8: Exit ranks. There is a rule (in Billboards charts) that albums cannot remain on the charts for more than 20 weeks (they are taken off the charts). This will influence the exit ranks.

Page 11 figure 8: Even though the fits are not as good as in figure 3, the tendency is very similar: the lines straighten out from being parabola-like. Should one interpret the worse fit as a qualitative, fundamental difference, or a quantitative one?

Page 12-13, sections (b) and (c) : what is the meaning of the hidden variable, h , in the context of music charts? In (c) it is stated that this is individual time horizons, but it is not very clear what this means. Please clarify this connection, otherwise the derivation in (b) does not apply.

Page 15, section 4, last paragraph: again, I think the relevance of this study to politics is not clear.

Author's Response to Decision Letter for (RSOS-190567.R0)

See Appendix A.

RSOS-190944.R0

Review form: Reviewer 1 (Zoltan Neda)

Is the manuscript scientifically sound in its present form?

Yes

Are the interpretations and conclusions justified by the results?

Yes

Is the language acceptable?

Yes

Do you have any ethical concerns with this paper?

No

Have you any concerns about statistical analyses in this paper?

No

Recommendation?

Accept with minor revision (please list in comments)

Comments to the Author(s)

I am satisfied with most of the answers provided in the new version of the manuscript, however the authors should still consider the following comments:

1. In the second half of one version of the manuscript references are not appearing in the PDF version, please double-check again what happened...And why there are two PDF-s of the manuscript in the submission?

2. Concerning the problem with $P(S)$ my point is:

$$P(S, S+dS) = P(s, s+ds) \rightarrow P(S)dS = p(s)ds$$

$$\text{so: } p(S) = p(s) \left(\frac{ds}{dS} \right) \Rightarrow P(S) = 1/S^a \quad \text{with } a = s_0/s_{\text{bar}} + 1.$$

this should be corrected in equation (3.1)

Review form: Reviewer 2

Is the manuscript scientifically sound in its present form?

Yes

Are the interpretations and conclusions justified by the results?

Yes

Is the language acceptable?

Yes

Do you have any ethical concerns with this paper?

No

Have you any concerns about statistical analyses in this paper?

No

Recommendation?

Accept with minor revision (please list in comments)

Comments to the Author(s)

Thanks very much for revising the paper in line with my comments. I think it makes some very valuable contributions, and I was glad I had the opportunity to review it.

Here are just some minor corrections.

(I still find the first paragraph of the abstract insufficiently motivated. But we can agree to disagree.)

page 3, line 11 "they allow US to study..."

line 20, "SUCH as the overall..."

line 23, "by a factor OF two or more"

page 5, line 57. I think they misunderstood my comment here. I meant replace "1.0" with "1", not with "1.". The "1." still makes me think it's data.

page 6, line 32, missing period before "For"

line 35 "as detailed in the Appendix" remove "out"

page 7, line 50, I would prefer "reversal" to "reversion"

page 14, line 55, democracies become inevitabLY unstable"

page 15, line 32, need to delete extra period between "information" and "increasingly"

lines 29 to 33. You lost me here. I thought according to your theory the power law was because of people basically being able to buy an album they want instantly. One time scale is taken out of the problem. So I guess "individuality" has been lost, in that there are now not different lengths of time for people to buy albums. And I didn't think that had anything to do with "personal preferences for musical variety". I think "growing irrelevance of individuality" begs for a romantic misinterpretation.

line 36. "information travels nowadays substantially faster". Of course, as you know, this is not literally true. Nothing travels faster than a radio wave. Can you say something more accurate at the risk of being less poetic?

Review form: Reviewer 3

Is the manuscript scientifically sound in its present form?

No

Are the interpretations and conclusions justified by the results?

No

Is the language acceptable?

Yes

Do you have any ethical concerns with this paper?

No

Have you any concerns about statistical analyses in this paper?

No

Recommendation?

Major revision is needed (please make suggestions in comments)

Comments to the Author(s)

The authors have made some changes in the manuscript, but I was somewhat disappointed that some of my central comments were brushed off. In particular, the authors still start their abstract by talking about democracies and elections. This has nothing to do with the work they have performed (unless they somehow demonstrate clearly that there is a connection). A careful analysis of chart behavior is what this paper presents, and this should not be over-interpreted as something significant for politics. Perhaps some connections could be drawn in the discussion section, but one cannot put this in the abstract. It is misleading.

With regards to political instabilities, the authors replied to my previous comment: "A reference is given, [37], where the theory is developed and everything is explained exactly. It is actually a very robust relation, time delay systems become necessarily unstable when the time delay (which is of the order of the electoral cycle) becomes larger than the instantaneous driving, opinion dynamics."

If the authors insist on making this connection, they should present the mathematical argument. What in their own work represents electoral cycles? What is the time delay? What are the equations? As the authors will agree with me, a mathematical statement that is true for one system of equations will not necessarily hold for another system of equations. So, when they say that everything is explained "exactly" in ref [37], it is necessary to explain how this is relevant to the current study, which does not discuss any dynamical system explicitly.

Decision letter (RSOS-190944.R0)

09-Jul-2019

Dear Dr Gros,

The Subject Editor assigned to your paper ("Five decades of US, UK, German and Dutch music charts show that cultural processes are accelerating") has now received comments from

reviewers. We would like you to revise your paper in accordance with the referee and Associate Editor suggestions which can be found below (not including confidential reports to the Editor). Please note this decision does not guarantee eventual acceptance.

Please submit a copy of your revised paper before 01-Aug-2019. Please note that the revision deadline will expire at 00.00am on this date. If we do not hear from you within this time then it will be assumed that the paper has been withdrawn. In exceptional circumstances, extensions may be possible if agreed with the Editorial Office in advance. We do not allow multiple rounds of revision so we urge you to make every effort to fully address all of the comments at this stage. If deemed necessary by the Editors, your manuscript will be sent back to one or more of the original reviewers for assessment. If the original reviewers are not available we may invite new reviewers.

When submitting your revised manuscript, you must respond to the comments made by the referees and upload a file "Response to Referees" in "Section 6 - File Upload". Please use this to document how you have responded to each of the comments, and the adjustments you have made. In order to expedite the processing of the revised manuscript, please be as specific as possible in your response.

- Ethics statement

- Data accessibility

<http://datadryad.org/submit?journalID=RSOS&manu=RSOS-190944>

- Competing interests

- Authors' contributions

- Acknowledgements

- Funding statement

on behalf of Marta Kwiatkowska (Subject Editor)
openscience@royalsociety.org

Reviewer comments to Author:

Reviewer: 1

Comments to the Author(s)

I am satisfied with most of the answers provided in the new version of the manuscript, however the authors should still consider the following comments:

1. In the second half of one version of the manuscript references are not appearing in the PDF version, please double-check again what happened...And why there are two PDF-s of the manuscript in the submission?

2. Concerning the problem with $P(S)$ my point is:

$$P(S, S+dS) = P(s, s+ds) \rightarrow P(S)dS = p(s)ds$$

$$\text{so: } p(S) = p(s) \left(\frac{ds}{dS} \right) \Rightarrow P(S) = 1/S^a \quad \text{with } a = s_0/s_{\text{bar}} + 1.$$

this should be corrected in equation (3.1)

Reviewer: 2

Comments to the Author(s)

Thanks very much for revising the paper in line with my comments. I think it makes some very valuable contributions, and I was glad I had the opportunity to review it. Here are just some minor corrections.

(I still find the first paragraph of the abstract insufficiently motivated. But we can agree to disagree.)

page 3, line 11 "they allow US to study..."

line 20, "SUCH as the overall..."

line 23, "by a factor OF two or more"

page 5, line 57. I think they misunderstood my comment here. I meant replace "1.0" with "1", not with "1.". The "1." still makes me think it's data.

page 6, line 32, missing period before "For"

line 35 "as detailed in the Appendix" remove "out"

page 7, line 50, I would prefer "reversal" to "reversion"

page 14, line 55, democracies become inevitabLY unstable"

page 15, line 32, need to delete extra period between "information" and "increasingly"

lines 29 to 33. You lost me here. I thought according to your theory the power law was because of people basically being able to buy an album they want instantly. One time scale is taken out of the problem. So I guess "individuality" has been lost, in that there are now not different lengths of time for people to buy albums. And I didn't think that had anything to do with "personal preferences for musical variety". I think "growing irrelevance of individuality" begs for a romantic misinterpretation.

line 36. "information travels nowadays substantially faster". Of course, as you know, this is not literally true. Nothing travels faster than a radio wave. Can you say something more accurate at the risk of being less poetic?

Reviewer: 3

Comments to the Author(s)

The authors have made some changes in the manuscript, but I was somewhat disappointed that some of my central comments were brushed off. In particular, the authors still start their abstract by talking about democracies and elections. This has nothing to do with the work they have performed (unless they somehow demonstrate clearly that there is a connection). A careful analysis of chart behavior is what this paper presents, and this should not be over-interpreted as something significant for politics. Perhaps some connections could be drawn in the discussion section, but one cannot put this in the abstract. It is misleading.

With regards to political instabilities, the authors replied to my previous comment: "A reference is given, [37], where the theory is developed and everything is explained exactly. It is actually a very robust relation, time delay systems become necessarily unstable when the time delay (which is of the order of the electoral cycle) becomes larger than the instantaneous driving, opinion dynamics."

If the authors insist on making this connection, they should present the mathematical argument. What in their own work represents electoral cycles? What is the time delay? What are the equations? As the authors will agree with me, a mathematical statement that is true for one

system of equations will not necessarily hold for another system of equations. So, when they say that everything is explained "exactly" in ref [37], it is necessary to explain how this is relevant to the current study, which does not discuss any dynamical system explicitly.

Author's Response to Decision Letter for (RSOS-190944.R0)

See Appendix B.

RSOS-190944.R1 (Revision)

Review form: Reviewer 3

Is the manuscript scientifically sound in its present form?

Yes

Are the interpretations and conclusions justified by the results?

Yes

Is the language acceptable?

Yes

Do you have any ethical concerns with this paper?

No

Have you any concerns about statistical analyses in this paper?

No

Recommendation?

Accept as is

Comments to the Author(s)

The authors have now adequately addressed my concerns.

Decision letter (RSOS-190944.R1)

29-Jul-2019

Dear Dr Gros,

I am pleased to inform you that your manuscript entitled "Five decades of US, UK, German and Dutch music charts show that cultural processes are accelerating" is now accepted for publication in Royal Society Open Science.

on behalf of Marta Kwiatkowska (Subject Editor)
openscience@royalsociety.org

Reviewer comments to Author:
Reviewer: 3

Comments to the Author(s)
The authors have now adequately addressed my concerns.

Follow Royal Society Publishing on Twitter: [@RSocPublishing](https://twitter.com/RSocPublishing)
Follow Royal Society Publishing on Facebook:
<https://www.facebook.com/RoyalSocietyPublishing.FanPage/>
Read Royal Society Publishing's blog: <https://blogs.royalsociety.org/publishing/>

Appendix A

Associate Editor Comments to Author:

- * It is perplexing that the authors have submitted to
- * Royal Society Open Science a manuscript that is evidently
- * in the cultural evolution space. This is not a comment on
- * the paper's quality (indeed, the reviewers have provided
- * extensive queries on this front), but rather a comment on
- * whether the authors have selected the correct journal for
- * their work. If the authors are able to satisfy the reviewers
- * that their required changes have been made, we will certainly
- * consider the paper for publication, but the authors might consider
- * a more field-specific journal for other papers in this realm.

We appreciate the concern. However several in-depth studies of music charts have been published previously by Royal Society Open Science, see Refs. [1] [8] [9] Could it be, that the policy of RSOS changed in the meantime? We ask with regard to future submissions.

Our analysis is furthermore motivated by complex systems theory, which we believe falls within the scope of RSOS.

Reviewer: 1

- * Comments to the Author(s)
- * The work presents evidences for the changes that are induced by an
- * accelerated selection process over the dynamics in the top-list of music
- * charts (Billboard, UK, German and Dutch). Convincing data is given
- * for visualizing the difference in the top-list dynamics and statistics
- * of music charts between * 1960-90 and 1990-2018. The accelerated rhythm
- * is nicely visible in the chart * diversity evolution. The distribution
- * time for the time periods in which an album persist in the top 40
- * positions has also visible differences between these two time-periods.
- * While in the period 1960-1990 one observes a lognormal distribution, in
- * the 1990-2018 period a scaling is observable (power-law trend). The
- * authors analyze also the differences in the evolutionary trends of
- * an album in the top-list. In the period 1960-90 mostly an increasing trend is
- * dominating (albums start at lower positions and climb towards the top...),
- * while in the 1990-2018 period a decreasing trend is observable (albums start
- * right in the top and slowly descends in the list). In contrast with albums,

* single songs do not have such a clear trends and statistics. The findings are
* explained in the context of information-theoretical approaches using entropy
* maximizing principles and the Weber-Fechner law. The manuscript is clearly
* presented and the topic is of broad interest. It is definitely a work that
* should be published, however before publication I would suggest the authors
* to take into account the following recommendations:

We thank the reviewer for his/her supporting comments.

We realized in the meantime, that the original Billboard sales charts have been retained after the 2014/15 update, albeit under a new name: Top Album Sales. We explain this now in the paragraph above section 2.(a). We updated all figures, using now the Billboard sales charts as our data source.

In Fig. 1/3 results from both ranking metrics are shown, from the official Top 200 and from the Top Album Sales.

We also added a new table, Table 1.

* 1. Why the authors focus only
* on the Western world. I assume that Asia is nowadays a leading market. Are
* there long-term charts available also for Japan, South Korea, etc...? Maybe
* some comments in this sense would help us.

This is correct. There are, however, very practical obstacles to non native speakers when it comes to downloading Asian data, such as being able to read Kanji. We hope that our manuscript will motivate subsequent studies.

* 2. The notation "log" is used for logarithm throughout the manuscript,
* without specifying if it is a ten based or "e" based logarithm. In some
* graphs, I had the feeling it is a ten based one, however in the equations
* I would tend that it is an "e" based one. Please use either the accepted
* notations: "lg" and "ln" or specify it.

In all formulas we have substituted \log by \ln . In the Figures \log_{10} has been used, as indicated now in the respective captions.

* 3. The term "unique album" could be a little confusing and not necessarily
* pointing to what the authors mean by that. Maybe some other expression or
* some explanations would be OK.

The number of 'unique albums' listed in a year is

the number of 'distinct albums'. We have changed the wording correspondingly.

- * 4. Equation 3.1 is incorrect. the P(S) scaling assuming the Weber-Fechner law
- * should be $P(S)=1/S^a$ with $a=s_0/s_{bar}-1$.

We actually cannot see why. To us 3.1 seems to be perfectly fine. Maybe a \log_{10} versus \ln issue?
We did not include the normalization factor, if this is the issue.

- * 5. In Equation 3.2 is not totally convincing for me the manner in which the
- * coupling is taken into account. I do understand that one needs a coupling
- * between the "h" and "s" variables, but why in the exponent (?) only for the
- * sake of analytic simplicity?

We added to the section, 3.(b), a short explanation of how maximum entropy distributions are derived, see the new Eq.(3.2).
It should be now evident that the coupling $a+\kappa h$ used in new 3.3 (old 3.2) represents that album lifetimes are of varying importance to people. We hope that this clarifies the issue.

Reviewer: 2

- * Comments to the Author(s)
- * I overall liked this paper and I thought the data was interesting and the
- * modeling in section 3 was good. My two main quibbles:

We thank the referee for his/her thoughts and comments.

We note that we added a new table, Table 1.

- * 1. The term "cultural time" implies to me that there is some underlying
- * phenomenon that is making all cultural things accelerate at about the same
- * rate. This seems pretty unlikely to me, but I guess you could demonstrate it
- * by looking at data from many different cultural domains. The paper only looks
- * at one cultural domain. And even if many domains were considered, wouldn't it
- * just be straightforward to say many cultural processes are speeding up. That
- * this is even true seems to be belied by Figure 1 where cultural time was
- * accelerating until 2014, and then switched to decelerating. I know that what
- * was measured changed in that year, but that is my point: if there was a

* "cultural time" that was accelerating, it wouldn't matter what we measured,
* we should still see it.

It is custom to use the terms cultural and political time in the literature examining the phenomenon of social acceleration. The referee is however correct that this would imply, strictly speaking, that a single time scale exists. This is of course not the case and has never been intended.

In order to clarify the terminology the term 'cultural processes' is now used in conjunction with 'distinct' cultural time scales.

We also changed the introduction at several places in order to make clear, that only a "set of culturally and sociologically relevant time scales" contribute to chart dynamics.

We would also like to point out that a previous study, [1], has used music charts (together with text-mining tools) to examine how musical styles have been changing. The author presented their analysis on the backdrop of cultural change in general, arguing that "Like any cultural artifact, ..., music is the result of a variational-selection process"

* 2. Given my skepticism about cultural time, I don't see what relevance the
* paper has to politics. But politics appear in the first sentence of the
* abstract. If you want to claim that political opinions are changing faster,
* then I think you really just have to look at political opinions. So in my
* opinion the paper would be improved by deleting "cultural time" and
* downplaying politics as a motivation.

It is of course correct, that the cultural time scales may be utterly independent of political time scales, at least as a matter of principles. For this to be true, political opinion formation and cultural processes would however need to be 100% decoupled, which seems to be unlikely.

Studies would suggest otherwise, namely that "cultural and political processes condition each other", for which references have been added in what is now the 'Discussion' section.

In any case, following the suggestion of the referee we have rewritten the conclusion, downplaying politics as

a motivation. We also tuned down its importance in the introduction, taking out the paragraph explaining the link to dynamical instabilities. We transferred this paragraph into a new 'Discussion' section (before the conclusions).

* smaller comments/questions:

- * page 1. in the abstract you use the phrase "unique phenomenon". How do you
- * know this is a unique phenomenon? Something going from a lognormal
- * distribution to a power law distribution has never happened before?

Yes, we have never seen that before. We have written an authoritative review on "Power laws and self-organized criticality in theory and nature", which is cited [16], but never encountered such a phenomenon. This is discussed now also in the conclusion. For complex systems theory, it is really very interesting.

- * page 2 "a unique observation" Again, can you say a little more about this?
- * It's never been observed before in cultural data? Or any kind of data? What
- * typically happens then.

Yes again. We reformulated and expanded the sentence in question.

- * page 2 sentence beginning with "It is postulated that human activities...."
- * was hard to understand. Maybe delete "to" after "viz".

Thanks. We reformulated the sentence substantially.

- * page 4. "1990th" -> "1990s"
- * in lines 56 and 57 replace 1.0 with 1. (When I see 1.0 I assumed it was from
- * data)

Yep, done.

- % page 5. "The trend to become more diverse was dramatically reversed..." I
- % don't think any real thing reversed trend. You just switched from looking
- % at one thing that was increasing to another thing that was decreasing. It
- % might be better just not to include the latter data.

We realized in the meantime, that the original Billboard

sales charts have been retained after the 2014/15 update, albeit under a new name: Top Album Sales. We explain this now in the paragraph above section 2.(a). We updated all figures, using now the Billboard sales charts as our data source. In Fig. 1/3 results from both ranking metrics are shown, from the official Top 200 and from the Top Album Sales.

- * page 6. "raising prominence" -> "rising prominence"
- * page 11. "give raise" -> "give rise"
- * page 13 "90th" -> "90s"

Thanks for pointing out these typos.

Reviewer: 3

- * Comments to the Author(s)
- * The paper "Five decades of US, UK, German and Dutch music charts show that cultural time is accelerating" provides an analysis of the song dynamics on top charts. Several interesting trends are found, including an accelerated turnover of songs on charts, the increased likelihood of songs entering as #1, and a change in the songs' lifetime distribution. I think this is an interesting study, but I find some of the interpretations made by the authors problematic. Also, below I give comments on how the analysis and presentation can be made clearer.

We thank the referee for carefully reading and interesting comments.

We realized, in the meantime, that the original Billboard sales charts have been retained after the 2014/15 update, albeit under a new name: Top Album Sales. We explain this now in the paragraph above section 2.(a). We updated all figures, using now the Billboard sales charts as our data source. In Fig. 1/3 results from both ranking metrics are shown, from the official Top 200 and from the Top Album Sales.

- * The main claim of the authors is that the observed statistical results that they obtain inform us about the changes in cultural time. Perhaps the authors should explain exactly what this means. The changes they observe are intuitively explained by the advent of the internet and information accessibility. Is this what they mean by cultural time? It is clear that information has become much more accessible across all cultural domains, and a detailed chart study is not necessary to claim this. Therefore, I would not

- * try to interpret the findings in the general "cultural time" framework, but
- * rather concentrate on the identifying the role of the technology on chart
- * behavior and music consumption. I would not interpret the results as
- * significant for democracy, politics, and other unrelated things. It is far
- * fetched and distracts the reader from the interesting results of the paper.

We agree that the term 'cultural time' is not specific enough, it may have caused misunderstandings. We changed it in the title, and throughout the paper, to 'time scale' and/or 'cultural process'. We believe that this term does not need further explanation. In our view it seems evident that the buildup of preferences for a given song is a cultural process, at least in part. This does not exclude sociological and economical components.

We took care to clearly state that our interpretations are interpretations and not conclusions. We went another time through the manuscript to make sure that our arguments were formulated carefully, correcting several instances.

Importantly, we have rewritten the conclusion. The link to opinion dynamics is now substantially less prominent.

Following the suggestion of the referee, we transferred the part of the introduction explaining the link to dynamical instabilities to a new 'Discussion' section (right before the conclusions).

The referee is of course correct that 'it is clear that information has become much more accessible across all cultural domains'. Studies showing quantitatively that cultural processes are accelerating are however rare, a key motivation for the present investigation.

- * Further, I think a more in-depth study of the chart rules should be
- * presented. There have been a lot of changes in the ways the charts were
- * formed, especially in the 90s and 2000s. These changes will obviously
- * affect the observables such as long lifespans, entry ranks, etc. This
- * information should be presented and used to explain the data.

A compendium has been added in the appendix.

- * Below are detailed comments.

* Abstract:

- * The abstract starts with statements about electoral cycles and modern democracies, but this seems somewhat removed from the main topic of the research. I suggest removing any mention of politics and motivate the research in a more direct way, by talking about the time scales if music preference changes.

This is a thoughtful comment. We believe however that it is important to point out that charts provide a rare instance of long-term consistently compiled data. Even though details changed, the fundamental criterion, success, stayed the same. Such kind of data will necessarily reverberate the effects of societal process at large. In effect we believe that it is a view of perspective which starting point to select.

* Page 3 line 20 - 25:

- * connecting some results from dynamics systems with political instabilities is very far fetched. It is not clear why these particular results should be applicable. At this point in the paper we don't know what the observations are exactly, and further it is unclear why an instability in a dynamics system has any relevance for "democracy in crisis". I suggest to either explain exactly how your results are connected with political instabilities (not in the intro, but in the discussion, after we know what the results are), or remove this part completely.

A reference is given, [37], where the theory is developed and everything is explained exactly. It is actually a very robust relation, time delay systems become necessarily unstable when the time delay (which is of the order of the electoral cycle) becomes larger than the instantaneous driving, opinion dynamics. We are sorry to disagree on this point. We corrected a typo in the section mentioned by the referee, which may have caused a misunderstanding.

* Page 3 lines 42-44:

- * "We believe that these empirical findings constitute quantitative evidence that the time scales determining cultural penetration and opinion formation processes have shortened substantially, in particular since the early 90s."
- * What is cultural penetration? Also, how do we know that the observations are caused by the shortening of the opinion formation, and not by any number of other reasons, much more mundane (e.g. the way the charts are created, the way music is consumed these days compared to the past, etc)?

We have reformulated the section in question in order to make clear that all we are doing is to propose an interpretation.

* Page 3, line 48: what does "a unique observation" refer to?

To our knowledge, it is the first time that the actual self-organizing process is being observed, and not just the end product. We have rewritten the sentence and added more explanations in the newly formulated conclusion.

* Page 3, line 54:

* what are "sensori stimuli" (a typo?) In general, this whole paragraph is not clear. It explains results that have not been presented yet, and it does so in a way that is not accessible to people not working directly in this type of problems. I suggest to move this to the discussion and provide more in-depth explanations.

Yes a typo, thanks for pointing it out (sensory). This section is part of the introduction, where one indicates which results and which theory will be presented later on, without all detail. The theory is explained later, in section 3.

* Page 4, sales vs airplay, e.g. line 48:

* ``In this study we concentrate mostly on sales, viz album charts." It appears that US top 100 charts contain information on both sales and airplay; please clarify.

It is explained in 2.(f) and also in the sentence above (" For airplay data, which are often included for single charts").

* Page 6, line 34:

* ``Note in this respect, that there are different routes on how to include streaming and song downloads" -- please clarify what this means.

A valid point. We updated the sentence and introduce a reference to the appendix, where it is now specified.

* Page 6, line 56:

* please remove "astonishing", and use a more quantitative description of the goodness of fit.

Done.

- * Page 7, line 55:
- * ``has been normalized such that the climbing time for albums entering at the
- * top is zero" -- please explain this normalization procedure.

The sentence in question has been reformulated.

- * Page 8, figure 5:
- * It could be informative in addition to show the probability distribution (a
- * histogram) of the initial rank of songs that ended up as #1, for example, for
- * 1985 and for 2015.

This is a good point. We examined the statistics, but there are only few #1 songs (maximally 52 in a year, normally much less). The distribution is hence so noisy that it does not tell anything. This is the reason we present average data, namely the number of weeks it takes to climb to the top, if that is ever reached.

Motivated by this suggestion, we added a new table, Table 1, in which a substantial amount of additional statistics for number one albums is given.

- * Pages 7-8: Number one entries. From Wikipedia:
- * ``During the 1990s, <...> accusations began to fly of chart manipulation
- * as labels would hold off on releasing a single until airplay was at its
- * absolute peak, thus prompting a top ten or, in some cases, a number one
- * debut. In many cases, a label would delete a single from its catalog
- * after only one week, thus allowing the song to enter the Hot 100, make
- * a high debut and then slowly decline in position as the one-time production
- * of the retail single sold out." -- can this have something to do with an
- * increased tendency to debut at number one position?

Thanks for the comment.

This applies, if the `accusations' are true, to the Billboard Hot 100 Single charts. We tried to find reliable sources for these `accusations', but could not find any. For the album charts no period of increase #1 entries can be found, it is a continuous transition. We do not find indications that these manipulation, if they existed, have had a lasting effect.

- * Page 8: Exit ranks.
- * There is a rule (in Billboards charts) that albums cannot remain
- * on the charts for more than 20 weeks (they are taken off the
- * charts). This will influence the exit ranks.

The rule was, from May 1991:

"Albums that were more than 18 months old and that had fallen below position 100 on the Billboard 200 were removed."

This rule was dropped December 2009.

This rule did in any case not affect the top 100 positions, that is the charts investigate in the present study.

* Page 11 figure 8:

- * Even though the fits are not as good as in figure 3, the tendency is very
- * similar: the lines straighten out from being parabola-like. Should one
- * interpret the worse fit as a qualitative, fundamental difference, or a
- * quantitative one?

Yes, the lines become roughly more straight, substantial deviations remain however. But we agree, a certain degree of crossover is to be expected, as the sales ranking of the corresponding albums should also be known to the program directors. We have added several sentences.

* Page 12-13, sections (b) and (c) :

- * what is the meaning of the hidden variable, h , in the context of music charts?
- * In (c) it is stated that this is individual time horizons, but it is not very
- * clear what this means. Please clarify this connection, otherwise the
- * derivation (b) does not apply.

A good point. We have added to the beginning of 3.(b) the standard deviation of a maximum entropy distribution. Lagrange parameters correspond in general to the relative weighing of the constraint. The internal parameter h expresses hence how important the constraint is for a given individual.

* Page 15, section 4, last paragraph:

- * again, I think the relevance of this study to politics is not clear.

Has been rewritten.

Appendix B

%%%

Reviewer: 1

%%%

- * Comments to the Author(s)
- * I am satisfied with most of the answers provided
- * in the new version of the manuscript, however the
- * authors should still consider the following comments:

- * 1. In the second half of one version of the manuscript
- * references are not appearing in the PDF version, please
- * double-check again what happened...And why there are two
- * PDF-s of the manuscript in the submission?

Thanks for pointing out these two issues. We had uploaded a pdf version and the TeX file, which seems to have created the problem.

- * 2. Concerning the problem with P(S) my point is:
- * $P(S, S+dS) = P(s, s+ds) \rightarrow P(S)dS = p(s)ds$
- * so: $p(S) = p(s) (ds/dS) \Rightarrow P(S) = 1/S^a$ with $a = s_0/s_{bar} + 1$.
- * this should be corrected in equation (3.1)

This is a very valid point, which we corrected. Thank you very much.

%%%

Reviewer: 2

%%%

- * Comments to the Author(s)
- * Thanks very much for revising the paper in line
- * with my comments. I think it makes some very valuable
- * contributions, and I was glad I had the opportunity
- * to review it. Here are just some minor corrections.

- * (I still find the first paragraph of the abstract
- * insufficiently motivated. But we can agree to disagree.)

We have rewritten the abstract, pushing the reference to politics to the back. It is just a remark that similar developments may occur in other domains.

- * page 3, line 11 "they allow US to study..."
- * line 20, "SUCH as the overall..."
- * line 23, "by a factor OF two or more"
- * page 5, line 57. I think they misunderstood my comment here.
- * I meant replace "1.0" with "1", not with "1.".
- * The "1." still makes me think it's data.
- * page 6, line 32, missing period before "For"
- * line 35 "as detailed in the Appendix" remove "out"
- * page 7, line 50, I would prefer "reversal" to "reversion"
- * page 14, line 55, democracies become inevitably unstable"
- * page 15, line 32, need to delete extra period between
- * "information" and "increasingly"

all done

- * lines 29 to 33. You lost me here. I thought according to your theory the
- * power law was because of people basically being able to buy an album
- * they want instantly. One time scale is taken out of the problem. So I
- * guess "individuality" has been lost, in that there are now not different

* lengths of time for people to buy albums. And I didn't think that had
* anything to do with "personal preferences for musical variety". I think
* "growing irrelevance of individuality" begs for a romantic
* misinterpretation.

Exactly, that is what was to be expressed.
The formulation was however somewhat misleading,
we corrected it.

* line 36. "information travels nowadays substantially
* faster". Of course, as you know, this is not literally true.
* Nothing travels faster than a radio wave. Can you say
* something more accurate at the risk of being less poetic?

True enough, reformulated. Thanks for the comments.

Reviewer: 3

* Comments to the Author(s)
* The authors have made some changes in the manuscript, but I was somewhat
* disappointed that some of my central comments were brushed off.

We are sorry if this has been the impression. We actually took
the comments very serious, restructuring the manuscript to a
substantial extent.

* In particular, the authors still start their abstract by talking about
* democracies and elections. This has nothing to do with the work they
* have performed (unless they somehow demonstrate clearly that there is a
* connection). A careful analysis of chart behavior is what this paper
* presents, and this should not be over-interpreted as something
* significant for politics. Perhaps some connections could be drawn in the
* discussion section, but one cannot put this in the abstract. It is
* misleading.

We recognize that our phrasing may not have reflected
our intentions to a full extend. We have rewritten the
abstract, shifting the connection to politics to the end.
I hope it is now clear, that we only suggest that our
findings, an acceleration of time scales for a cultural
process, could be seen as an incentive to study the
evolution of political time. Nothing more.

* With regards to political instabilities, the authors replied to my
* previous comment: "A reference is given, [37], where the theory is
* developed and everything is explained exactly. It is actually a very
* robust relation, time delay systems become necessarily unstable when the
* time delay (which is of the order of the electoral cycle) becomes
* larger than the instantaneous driving, opinion dynamics."

* If the authors insist on making this connection, they should present the
* mathematical argument. What in their own work represents electoral
* cycles? What is the time delay? What are the equations? As the authors
* will agree with me, a mathematical statement that is true for one system
* of equations will not necessarily hold for another system of equations.
* So, when they say that everything is explained "exactly" in ref [37], it
* is necessary to explain how this is relevant to the current study, which
* does not discuss any dynamical system explicitly.

The connection to political issues is all in the discussion section.
We believe that a finding in one area, chart dynamics, motivates
to investigate whether a similar phenomenon occurs in other domains,

such as politics. This is in essence all we suggest. We do not claim that the dynamical stability of democracies is of any relevance to the present study or that our findings prove in any way that political time accelerates equally. It is however a hint that this may be happening. We hope that our phrasing makes this clear and that the referee may agree with it.

The reason that we do not provide the equations developed in Ref [37] is that we agree with the referee, they are not of relevance for the present study. All we want to say that [37] is a motivation to study the same topic of the present paper, the long-term evolution of time scales, also in other domains. We believe that this is an additional angle of view that may of interest to some of the readers.